# Van Krevelen diagrams based on machine learning visualize feedstock-product relationships in thermal conversion processes

Shule Wang [1,2,14], Yiying Wang[3,14], Ziyi Shi[4,14], Kang Sun[1,2,14], Yuming Wen [3✉], Lukasz Niedzwiecki [5,6], Ruming Pan [7,8], Yongdong Xu[9], Ilman Nuran Zaini[4], Katarzyna Jagodzińska[4], Christian Aragon-Briceno[10], Chuchu Tang[11], Thossaporn Onsree[12], Nakorn Tippayawong[13], Halina Pawlak-Kruczek[5], Pär Göran Jönsson[4], Weihong Yang[4], Jianchun Jiang [1,2✉], Sibudjing Kawi [3✉] & Chi-Hwa Wang[3✉]

Feedstock properties play a crucial role in thermal conversion processes, where understanding the influence of these properties on treatment performance is essential for optimizing both feedstock selection and the overall process. In this study, a series of van Krevelen diagrams were generated to illustrate the impact of H/C and O/C ratios of feedstock on the products obtained from six commonly used thermal conversion techniques: torrefaction, hydrothermal carbonization, hydrothermal liquefaction, hydrothermal gasification, pyrolysis, and gasification. Machine learning methods were employed, utilizing data, methods, and results from corresponding studies in this field. Furthermore, the reliability of the constructed van Krevelen diagrams was analyzed to assess their dependability. The van Krevelen diagrams developed in this work systematically provide visual representations of the relationships between feedstock and products in thermal conversion processes, thereby aiding in optimizing the selection of feedstock and the choice of thermal conversion technique.

[1] Jiangsu Province Key Laboratory of Biomass Energy and Materials, National Engineering Laboratory for Biomass Chemical Utilization, Institute of Chemical Industry of Forest Products, Chinese Academy of Forestry (CAF), 210042 Nanjing, China. [2] Jiangsu Co-Innovation Center for Efficient Processing and Utilization of Forest Resources, College of Chemical Engineering, Nanjing Forestry University, Longpan Road 159, 210037 Nanjing, China. [3] Department of Chemical and Biomolecular Engineering, National University of Singapore, 4 Engineering Drive 4, Singapore 117585, Singapore. [4] Department of Materials Science and Engineering, KTH Royal Institute of Technology, SE-100 44 Stockholm, Sweden. [5] Department of Energy Conversion Engineering, Wroclaw University of Science and Technology, 27 wybrzeże Stanisława Wyspiańskiego st. 50-370, Wroclaw, Poland. [6] Energy Research Centre, Centre for Energy and Environmental Technologies, VŠB-Technical University of Ostrava, 708 00 Ostrava, Poruba, Czech Republic. [7] School of Energy Science and Engineering, Harbin Institute of Technology, 150001 Harbin, China. [8] Institut de Mécanique des Fluides de Toulouse (IMFT) - Université de Toulouse, CNRS-INPT-UPS, 31400 Toulouse, France. [9] Laboratory of Environment-Enhancing Energy (E2E), Key Laboratory of Agricultural Engineering in Structure and Environment of Ministry of Agriculture and Rural Affairs, China Agricultural University, 100083 Beijing, China. [10] Department of Industry and Energy, CIRCE-Research Centre for Energy Resources and Consumption, 50018 Zaragoza, Spain. [11] Faculty of Creative Arts, University of Malaya, 50603 Kuala Lumpur, Malaysia. [12] Department of Chemical Engineering, University of South Carolina, 301 Main St, Columbia, SC 29208, USA. [13] Department of Mechanical Engineering, Chiang Mai University, 239 Huay Kaew Rd., Muang District, Chiang Mai 50200, Thailand. [14]These authors contributed equally: Shule Wang, Yiying Wang, Ziyi Shi, Kang Sun. ✉email: yuming@nus.edu.sg; jiangjc@caf.ac.cn; chekawis@nus.edu.sg; chewch@nus.edu.sg

The thermal conversion process, also known as the thermochemical process, has been widely developed and utilized for treating waste/biomass during recent decades[1,2]. These thermal conversion techniques offer options for power generation, fuel production, and chemical synthesis from different feedstocks[3]. Combustion, as a prevalent thermal conversion process, has been extensively studied and well-understood in terms of energy production and resulting by-products[4]. Emerging techniques such as torrefaction, pyrolysis, gasification, and hydrothermal liquefaction (HTL) have received more attention and present more complex performance dynamics[5,6].

The feedstock properties used in thermal conversion processes play a critical role in their performance. As a result, numerous studies have been conducted to investigate the performance of different thermal conversion processes using various raw materials[7], requiring significant research resources. The complexity of feedstock composition hinders researchers from identifying general principles through experimental studies. Having a simple model or guidance tool that can provide a preliminary estimation of thermal conversion process behavior based on feedstock properties would facilitate and accelerate the research and decision-making processes. Such a tool could indicate the relationship between the feedstock properties and the quantity and quality of products obtained from thermal conversion processes. Additionally, it could serve two practical objectives: guiding the selection of a suitable thermal conversion technique for a specific feedstock and assisting in determining the optimal feedstock or blend for a particular thermal conversion technique.

The van Krevelen diagram, introduced by Dirk Willem van Krevelen in 1950[8], displays the atomic ratios of H/C and O/C and was originally used to illustrate humification and coal formation processes visually[9]. Over time, it has been recognized as a useful tool for estimating main compound categories and reflecting their calorific values[10]. Consequently, its application has expanded beyond coal, denoting relevant properties of diverse materials, including biomass, biodegradable waste, and various chemicals, both pre- and post-reactions[11–19].

In the field of thermal conversion, the van Krevelen diagram has been widely used to intuitively indicate differences in H/C and O/C ratios among feedstocks and products in processes such as torrefaction[20], hydrothermal carbonization[21], pyrolysis[22], and gasification[23]. This application provides a unique way to visually illustrate the directions of not only thermal conversions but also other chemical reactions[24–28]. However, real feedstocks, such as biomass and biodegradable waste, are typically mixtures, implying that numerous reactions can occur during the thermal conversion process. Consequently, previous investigations using the van Krevelen diagram to understand the directions of several specific reactions can be challenging to apply to the analysis of mixtures. On the other hand, although there have been studies using the van Krevelen diagram to illustrate the thermal conversion reactions of real biomass and biodegradable wastes such as algae[29], lignocellulosic biomass[30], and digestate[31], typically only single or a few cases are reported in each study. Therefore, there is interest in addressing these gaps and creating van Krevelen diagrams that better reflect the real-world applications of different thermal conversion techniques.

Machine learning (ML) has become widely used in various fields[32], including constructing models for thermal conversion processes[33,34]. In most of the reported ML studies of thermal processes, the constructed ML model can predict the output from given input parameters with a coefficient of determination ($R^2$) higher than 0.8[33]. One ML interpretation method, the partial dependence plot, can be used to evaluate the marginal effects of selected input variables on the output value[35]. By using the H/C and O/C ratios of feedstocks as input parameters for an ML

model and plotting the two-way partial dependence of these input variables on the output value, a three-dimensional van Krevelen diagram can be created. It will be promising to use the ML method to construct the van Krevelen diagram: using the database yielded from experiments with mixture feedstock will give insight into the corresponding thermal process to treat real feedstock.

In this analysis, we create a series of van Krevelen diagrams to illustrate the relationship between the feedstock and its thermal conversion products through ML analysis of eight corresponding ML studies (Fig. 1). We analyze the constructed van Krevelen diagrams theoretically and propose suggestions for applying different thermal conversion processes to treat waste/biomass based on the diagrams generated. The reliabilities of the constructed diagrams are discussed. The study demonstrates that the constructed van Krevelen diagrams can effectively represent the feedstock-product relationship of part of the thermal conversion processes and can serve as an important reference for decision-making in different applications.

## Results and discussion

**Construction of van Krevelen diagram.** Figure 1 depicts the process of constructing and analyzing a van Krevelen diagram, using the yield of pyrolysis oil as an example. The database used for constructing the diagram is sourced from a precious study[36] and revised accordingly. Among the eight referenced studies, the random forest method has been implemented most frequently, with a testing $R^2$ value greater than 0.75 (as shown in Fig. 1). To establish a set of general diagrams for different thermal conversion processes, we consistently employ the random forest method verified its performance using the Leave One Out method (90% train data and 10% test data). The model is constructed using the scikit-learn 0.23.1 library in a Python 3 environment and details are given in Supplementary Note 4.

To generate the van Krevelen diagram, a two-way partial dependence analysis is performed based on the constructed model, focusing on the H/C and O/C ratios. The reliability of the produced diagram is highest within the ranges where the training data is most abundant. The dataset was initially collected from reported experimental works. In these experiments, researchers often explored various thermal conversion process parameters for a single feedstock. Therefore, plotting the H/C and O/C ranges for the collected data cannot accurately represent the true density of the dataset. To address this, we apply the kernel density of the training data to determine the appropriate ranges for the van Krevelen diagrams. The kernel density plots created for all datasets are given in Supplementary Figs. 7–10.

During the analysis, all other input parameters are set to their mean values from the training dataset. The resulting two-way partial dependence plots represent the predicted outcomes under specific input conditions. To obtain a van Krevelen diagram that can represent the relationship between feedstock and product more generally, smoothing is applied to the original two-way partial dependence plots. The detailed setting of smoothing is given in Supplementary Fig. 5.

The thermal conversion reaction models for individual feedstocks, such as cellulose, have been well developed. However, in real-world scenarios, raw materials are typically mixtures. This study demonstrates that machine learning methods can be utilized to capture the complexities of reactions involving mixed feedstocks (Supplementary Note 1). It's important to note that elements in the feedstock other than C, H, and O can significantly influence the properties of the final product, such as its ash content (Supplementary Note 2). However, this study focuses solely on the C, H, and O contents, as they are the most abundant

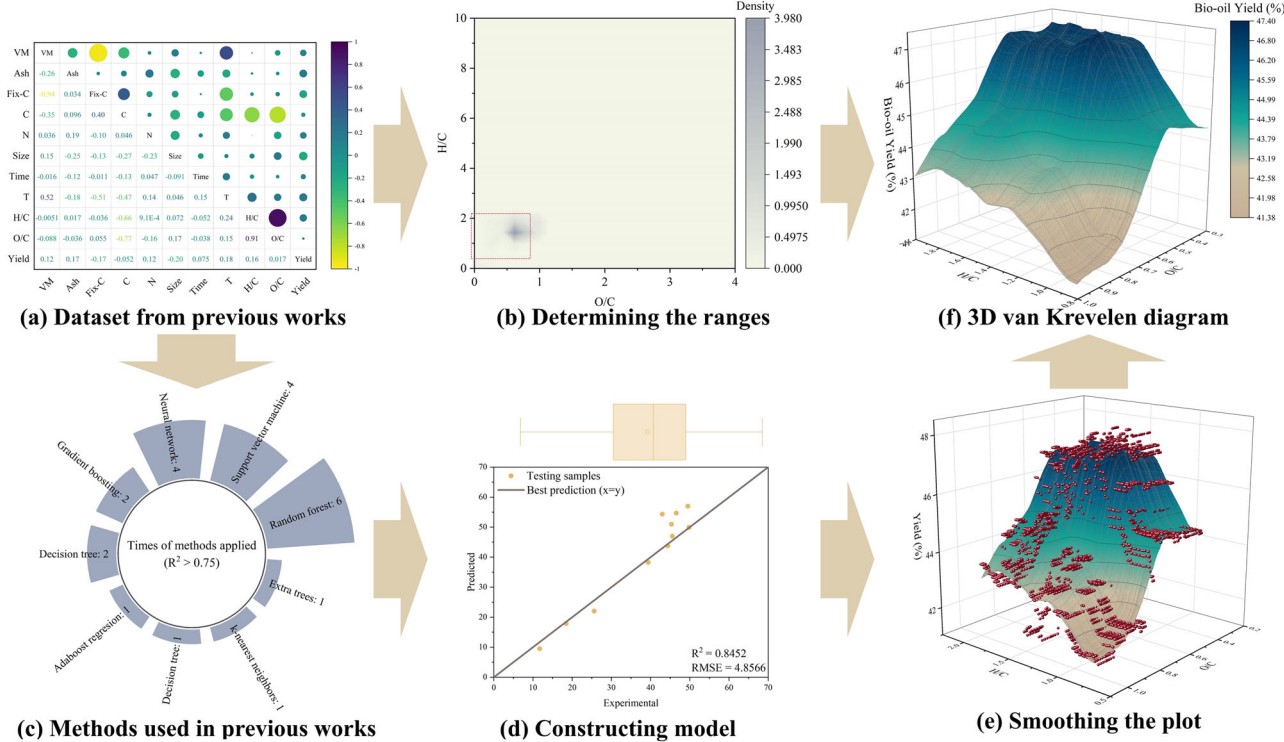

**Fig. 1 Schematic of the construction and analysis of van Krevelen diagrams. Using the yield of pyrolysis oil as an example. a** Pearson correlation of the raw dataset[36]. **b** Kernel density diagram for determination of the range of the predicting area. **c** The ML methods used in collected literature. **d** The fitting diagram of trained model. **e** Plot smoothing of the 2D-partial dependence plot which used H/C and O/C as axis. **f** The ascended 3D van Krevelen diagram.

elements in organic feedstocks. Considering the intricate nature of feedstocks, the complexity further intensifies when aiming to produce chemical products (Supplementary Note 3). Therefore, this study focuses on analyzing the fuel properties of thermal conversion products.

**Torrefaction**. During torrefaction, partial devolatilization occurs, leading to a decrease in volatile matter content[20,37]. This process makes the torrefied material more similar to coal compared to unprocessed biomass. Figure 2 demonstrates a generally negative correlation between the H/C and O/C ratios of the feedstock and the solid yield from biomass torrefaction, with a stronger correlation observed for the H/C ratio. The trend aligns with findings in the literature[37,38] and can be attributed to the composition of lignocellulosic biomass, particularly the hemicellulose content. Hemicellulose and cellulose have higher H/C and O/C ratios compared to lignin[37]. Thus, as the hemicellulose content increases in the raw biomass, the H/C and O/C ratios of the biomass also increase. The main mechanisms involved in biomass torrefaction are dehydration and decarboxylation[39]. Hemicellulose, which contains abundant hydroxyl groups, undergoes degradation during torrefaction through dehydration and the breaking of weak linkages between small substituents and the main polymer chains[40]. Generally, due to its lower thermal stability compared to cellulose and lignin[41], hemicellulose decomposition prevails at lower temperatures, such as in torrefaction. Therefore, a higher hemicellulose content results in more intense devolatilization and, ultimately, a lower solid yield from the process for the same severity of the torrefaction process[42].

**Hydrothermal processes**
*Hydrothermal carbonization (HTC)*. The reactions during HTC involve hydrolysis, dehydration, decarboxylation, condensation,

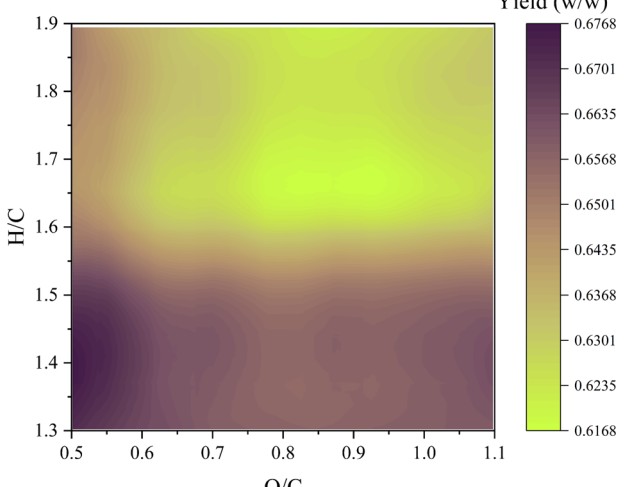

**Fig. 2 Van Krevelen diagram for torrefaction product. The diagram is constructed based on previous work by Onsree et al.[70].** The feedstock investigated is lignocellulosic biomass, including agricultural and forestry residues, as well as energy crops. The *x*-axis and *y*-axis represent the O/C and H/C of feedstock, respectively, while the *z*-axis indicates the yield of the product.

polymerization, aromatization, and condensation, among others[43]. Dehydration and decarboxylation are the primary mechanisms in HTC[44]. Consequently, higher H/C and O/C ratios in the feedstock result in higher H/C and O/C ratios in the produced hydrochar, as shown in the HTC H/C and H/O diagrams (Fig. 3).

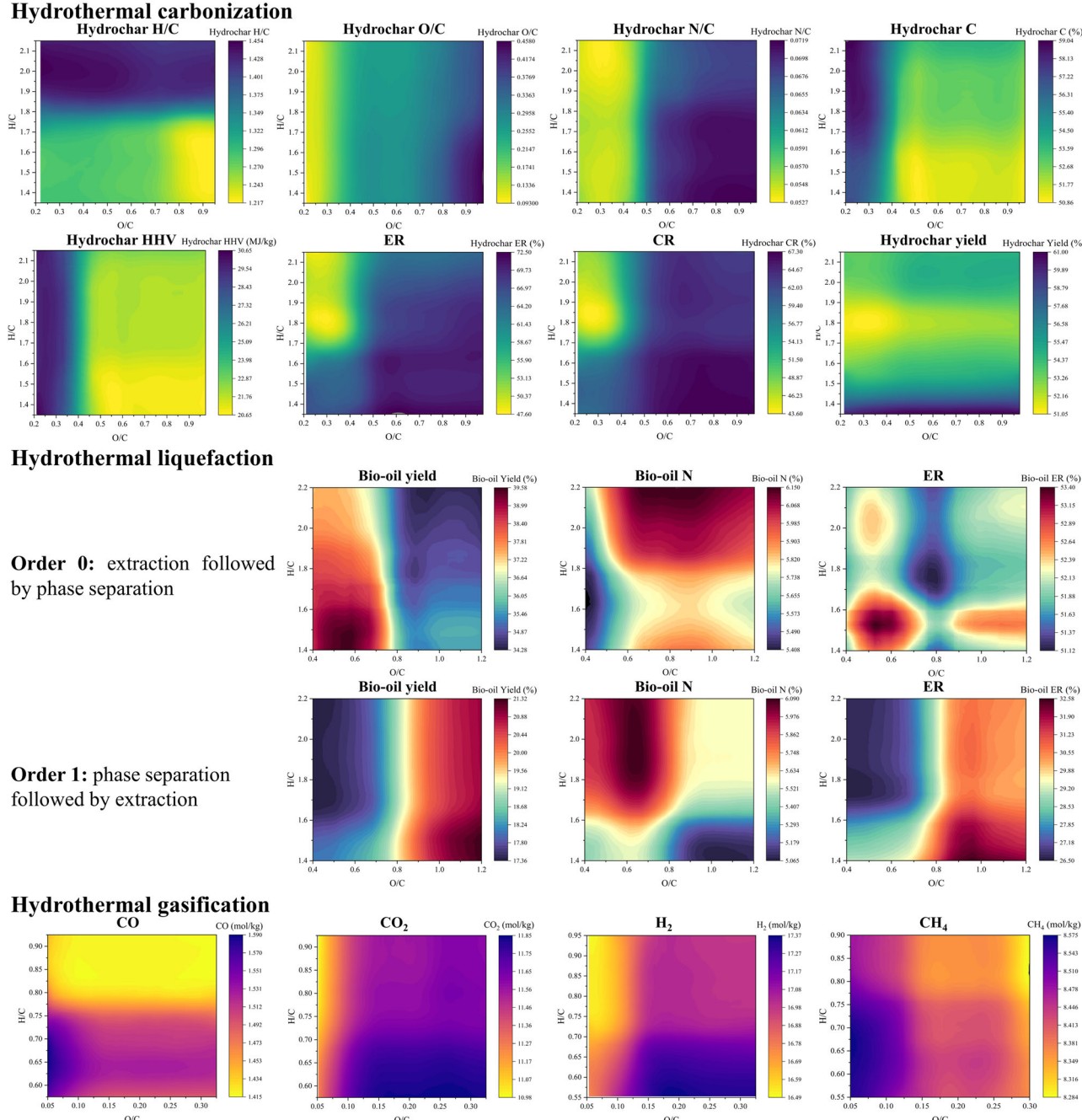

**Fig. 3 Van Krevelen diagrams for hydrothermal carbonization (HTC), hydrothermal liquefaction (HTL), and hydrothermal gasification (HTG) processes.** The HTC diagrams are established based on previous work by Li et al.[71] using biodegradable waste such as sewage sludge, food waste, and manure as the investigated feedstock. The HTL diagrams are established based on previous work by Li et al.[55] using biomass and biodegradable waste, including algae, sludge, food waste, and manure as the investigated feedstock. The HTG diagrams are established based on previous work by Liu et al.[72] using coal as the investigated feedstock. HHV higher heating value, ER energy recovery, CR carbon recovery.

The training data for HTC includes various types of biodegradable wastes. For food waste, different intermediates are produced during hydrolysis, with amino acids being the main intermediate after protein hydrolysis[44]. These intermediates serve as substrates for producing heterocyclic compounds[45], particularly N-containing ring compounds[46], through Maillard reactions[45,46]. This explains why using feedstocks located in regions with relatively high O/C ratios and relatively low H/C ratios results in the production of hydrochar with a higher N/C ratio. Additionally, sewage sludge typically contains significant

amounts of undigested proteins and extracellular polymeric substances from microbial aggregates[47,48].

In most empirical formulas, the HHV value is primarily determined by the C, H, and O contents. The C and H contents positively correlate with HHV value, while the O content has a negative correlation[49]. The HTC HHV diagram aligns with these trends. The results in the HTC C diagram exhibit a similar trend to the results in the HTC HHV diagram, possibly due to the higher carbonization degree of hydrochar, which increases the influence of the C content on the HHV value. Consequently, the

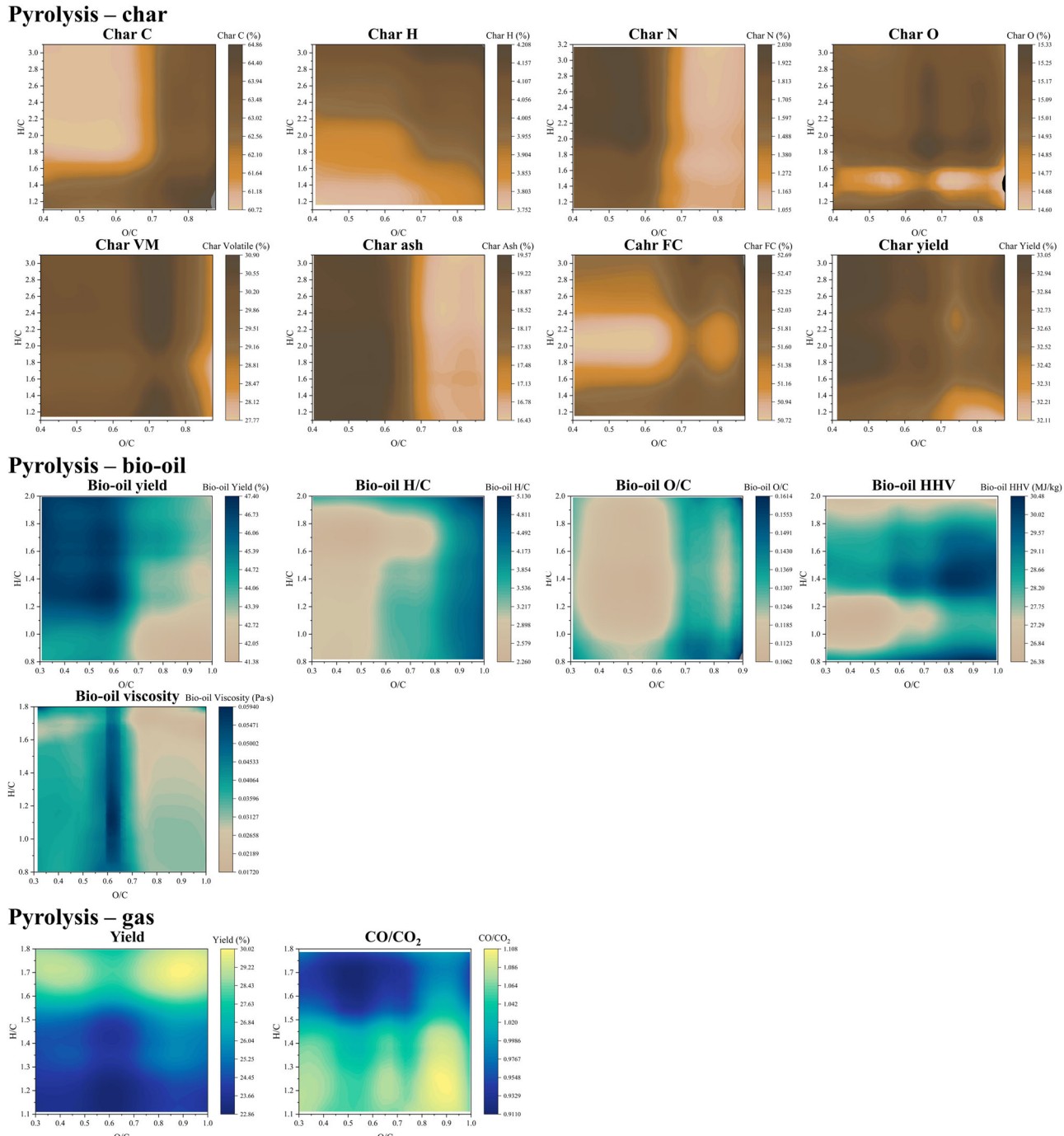

**Fig. 4 Van Krevelen diagrams for the products of char, bio-oil, and gas yielded from pyrolysis. The char diagrams are established based on the previous work by Li et al.[73].** The investigated feedstock is lignocellulosic biomass, including corncob, rice husk, sawdust, wood, etc. The bio-oil diagrams are established based on the previous work by Zhang et al.[36]. The investigated feedstock is biomass, including seed, rice husk, algae, etc. The pyro-gas diagrams are established based on the previous work by Tang et al.[74]. The investigated feedstock is biomass, including agriculture and forest waste, algae, etc. VM volatile matter, FC fix carbon.

HTC HHV diagram and the HTC C diagram share similarities. C content is important in predicting energy yields, as evidenced by the pattern observed in the HTC ER diagram, which mirrors the pattern in the HTC CR diagram. This similarity is not surprising, as the equations for calculating ER (Eq. 2) and CR (Eq. 3) depend linearly on the mass yield Eq. 1.

Lignin has a lower H/C ratio (1.14) compared to cellulose (1.67) and hemicellulose (1.60)[50]. Therefore, a lower H/C ratio in lignocellulosic biodegradable waste indicates a relatively higher

lignin content. Lignin exhibits better thermal stability than cellulose and hemicellulose, which is reflected in the HTC Yield diagram: the lower the H/C ratio, the lower the yield. Similarly, the HTC C diagram illustrates that a lower H/C ratio results in lower C content in the hydrochar. The hydrochar produced from lignin has a relatively lower C content than that produced from cellulose and hemicellulose[51]. Hence, feedstock with a higher lignin content will have a lower H/C ratio and produce hydrochar with lower C content.

**Hydrothermal liquefaction (HTL)**. The composition of the feedstock elements and compounds, as well as the order of product separation, play a crucial role in HTL[52]. Some small molecular compounds are considered as being light biocrude oil and can be extracted using organic solvents such as petroleum ether, dichloromethane, acetone, and ethyl acetate[53,54]. In the original work[55], order 0 represents bio-oil extraction followed by phase separation, while order 1 represents the reverse process. Consequently, bio-oil obtained using order 1 mainly contains heavy organics, whereas bio-oil obtained using order 0 contains both light and heavy organics. As depicted in the HTL yield diagrams, the bio-oil yield and ER values of order 1 are relatively higher compared to those of order 0.

Both orders in HTL yield diagrams indicate that a low H/C in the feedstock favors bio-oil yield. However, the HTL yield diagram for order 0 shows that a low O/C contributes to a high bio-oil yield, while the opposite trend is observed in the HTL yield diagram for order 1. The main organics in biodegradable waste are carbohydrates, lipids, and proteins. Lipids have the relatively lowest O/C ratios[56] and the highest HTL bio-oil yield rates among these organics[57]. Furthermore, the HTL of lipids produces more light organics than that of lignocelluloses and proteins[58,59]. Consequently, a higher O/C ratio in the feedstock leads to a higher yield of heavy organics and a lower yield of light organics in the bio-oil.

*Hydrothermal gasification (HTG)*. $N_2$, $CO_2$, $H_2$, $CH_4$, and CO are the main gases generated from HTG[60]. HTG reactions are complex[61], but the diagrams can be explained through the main reactions. The HTG diagrams for $CO_2$ and $H_2$ exhibit similar patterns because both gases are products of the reaction $C + 2H_2O \rightleftharpoons CO_2 + 2H_2$. Moreover, the $H_2$ and $CH_4$ diagrams exhibit a reverse pattern, which can be attributed to the reaction $C + 2H_2 \rightleftharpoons CH_4$.

A lower H/C ratio of a feedstock tends to result in a higher yield of CO, $CO_2$, and $CH_4$, as shown in the corresponding HTG diagrams. This can be explained by the requirement of C for the production of C-containing gases. Regarding the feedstock's O/C ratio, a lower O/C ratio increases the yields of CO and $CH_4$ while decreasing the yield of $CO_2$. A lower O/C ratio in the feedstock reduces the abundance of O and favors $CH_4$ production, while incomplete reactions produce CO instead of $CO_2$.

**Pyrolysis**

*Pyrolysis-char*. The pyrolysis-char H diagram demonstrates an apparent positive correlation between the H/C ratio of feedstock and the H content of char, as shown in Fig. 4. This can be attributed to feedstock's higher initial hydrogen content, resulting in more hydrogen residues in the solid products. However, the patterns observed in the other seven pyrolysis-char diagrams are challenging to explain. These results will be discussed in the section on "Reliability analysis" in Supplementary Note 6.

*Pyrolysis-bio-oil*. Pyrolysis-bio-oil Yield diagram shows that a higher H/C ratio and lower O/C ratio lead to increased bio-oil yields. Caprariis et al.[62] observed that the use of oak wood with a high H/C ratio could generate a higher oil yield compared to the use of natural hay and walnut shell with low H/C ratios, which is in agreement with the result in the diagram.

The H/C and O/C diagrams for Pyrolysis-bio-oil highlight that the O/C ratio of the biomass predominantly influences the H/C and O/C ratios in the bio-oil. Generally, an elevation in the biomass's O/C ratio correspondingly increases the H/C and O/C ratios in the bio-oil. Cellulose and hemicellulose exhibit O/C ratios of 0.83 and 0.80, respectively. These values are noticeably

higher than lignin's O/C ratio, which stands at 0.35[50]. Notably, bio-oil derived from the pyrolysis of cellulose and hemicellulose (xylan) demonstrated a superior H/C ratio compared to that from lignin pyrolysis[63]. As a consequence, a higher O/C ratio in biomass, indicating a reduced lignin content, translates to a heightened H/C ratio in the resultant bio-oil.

The Pyrolysis-bio-oil HHV diagram shows that the bio-oil tends to have a relatively low HHV value when derived from biomass with low H/C (0.90–1.25) and O/C (0.30–0.75) ratios. However, the HHV of bio-oil reaches its maximum value for biomass with medium H/C (1.30–1.50) and high O/C (0.75–1.00) ratios. It should be noted that higher HHV values correspond to a higher economic value of bio-oil, as its combustion can generate more heat, providing meaningful guidance for the commercial applications of bio-oils derived from different types of biomasses.

The viscosity of bio-oil is more sensitive to the O/C ratio in the biomass, as is shown in the bio-oil Viscosity diagram. The bio-oil derived from biomass with O/C ratios ranging from 0.60 to 0.65 has a high viscosity, suggesting that the bio-oil has a high average molecular mass[64]. Hence, biomass with an O/C ratio in this range is most recommended for producing bio-oil for lubricating oil applications. The pyrolysis of biomass with high H/C and O/C ratios can decrease the viscosity in the bio-oil, which agrees with the experimental observations by Fahmi et al.[64].

*Pyrolysis-gas*. Higher H/C and O/C values of feedstocks represent a higher potential for carbonization, resulting in a higher yield of pyro-gas, as shown by the Pyrolysis-gas Yield diagram. The main gases produced from biomass pyrolysis are $CO_2$, CO, $H_2$, and $CH_4$[65]. Except for $CO_2$, the other three gases can be used as fuel. Therefore, it is desirable to minimize the yield of $CO_2$ when considering the pyro-gas. The production of $CO_2$ and CO is highly correlated. The $CO/CO_2$ ratio can be an important parameter for evaluating the fuel quality of a pyro-gas. The pyrolysis-gas $CO/CO_2$ diagram shows that the $CO/CO_2$ ratio is mainly determined by the feedstock's H/C ratio: the yield of CO can be higher than that of $CO_2$ when the feedstock's H/C is lower than approximately 1.5. The HTG CO diagram also shows a trend that a higher CO yield is correlated to a lower feedstock H/C, which can be explained by the high C content supplied from the feedstock.

**Gasification**. The Gasification Yield diagram suggests lower O/C and higher H/C values lead to higher syngas yields, as illustrated in Fig. 5. The diagram shows the highest syngas yield when O/C < 0.35 and H/C > 1.70, which typically represents the fuel compositions of plastic-rich waste materials[66]. Similar findings have been reported in previous studies on biomass and waste gasification. For example, Arena et al.[67] studied the gasification of five different types of waste using a pilot-scale bubbling fluidized bed gasifier. They found that the waste fraction containing mostly plastics generated a higher syngas yield compared to the gasification of packaging waste, which had a lower plastic and higher lignocellulosic fraction.

The yield of CO in the syngas is directly proportional to the O/C content of the feedstock and does not vary significantly with different H/C values, as is shown in the Gasification CO diagram. Gasification of feedstock containing oxygenated compounds (with higher O/C ratios) produces more CO[68]. On the other hand, the relationship between the $CO_2$ yield and the H/C and O/C ratios of the feedstock is more complex, as depicted in the Gasification $CO_2$ diagram. To minimize the generation of $CO_2$ in the syngas, feedstock with O/C ratios between approximately 0.35 and 0.45 should be used, while the production of $CO_2$ is likely to increase when the O/C value is outside that range.

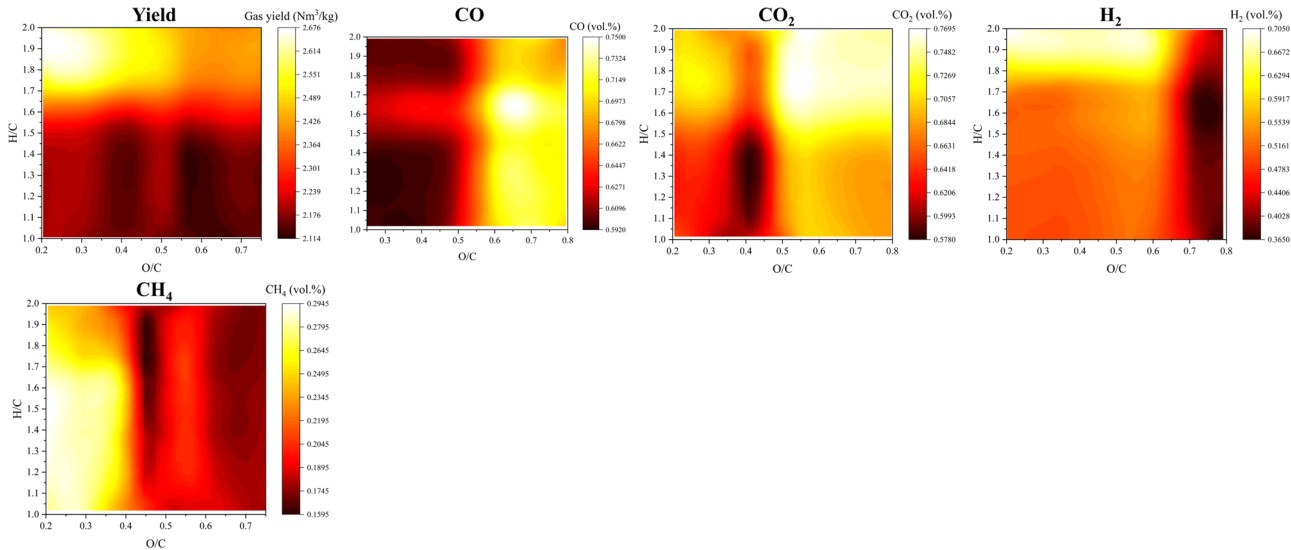

**Fig. 5 Van Krevelen diagrams for the gas products of gasification. The gasification diagrams are established based on the previous work by Serrano et al.[75].** The investigated feedstock is lignocellulosic biomass and plastics.

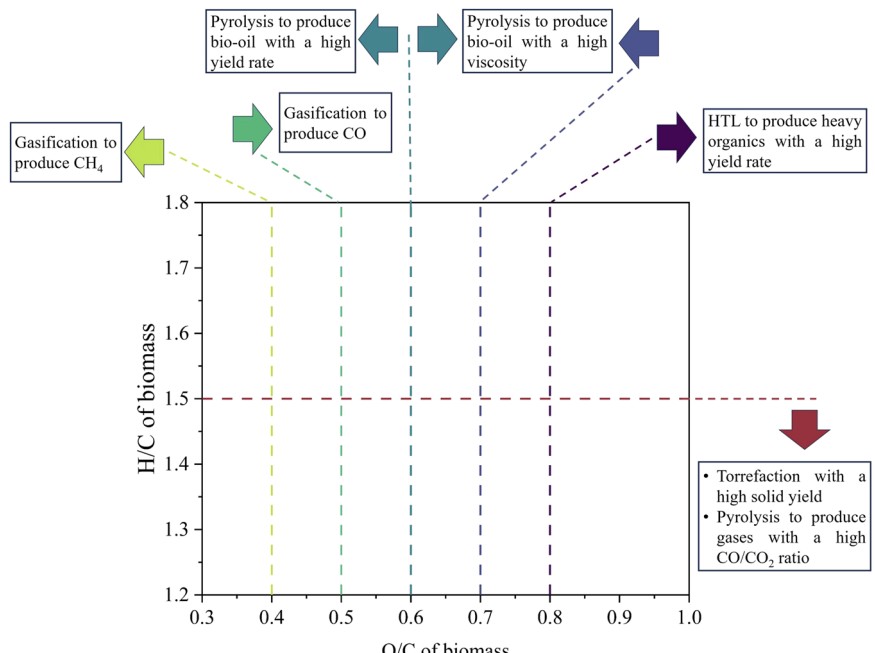

**Fig. 6 Summary of recommendations for the thermal conversion of biomass to achieve different targets, illustrated using a Van Krevelen diagram.** Only models created from datasets containing biomass feedstock are included.

In the Gasification $H_2$ diagram, at O/C < 0.6, higher H/C values in biomass result in higher $H_2$ yields. However, at O/C > 0.6, higher H/C values do not always correspond to higher $H_2$ yield. Instead, the value decreases significantly and remains relatively constant at different H/C values. This observation aligns with previous studies on fluidized bed gasification of biomass and plastic waste, which have shown that increasing the H/C ratio by increasing the plastic fraction in biomass-plastic fuel mixtures generally leads to a higher $H_2$ generation[68].

The Gasification $CH_4$ diagram follows a similar trend to the gasification diagram of $H_2$, reaching its higher level when the feedstock has lower O/C values. The production of $CH_4$ during waste gasification is linked to the fundamental components of the waste materials. Long-chain hydrocarbon compounds in plastics undergo cracking and reforming reactions during gasification, resulting in the formation of lighter hydrocarbon gases such as $CH_4$ and $C_2H_4$[69]. This process explains why the yield of $CH_4$ is higher during the gasification of high H/C feedstock, such as plastics.

**Summary of biomass thermal conversions.** A brief summary is provided in this section, outlining recommendations for the thermal conversion of biomass with varying H/C and O/C ratios to achieve different target products. Notably, ML models for torrefaction, HTL, pyrolysis-bio-oil, pyrolysis-gas, and gasification were built using datasets from biomass feedstock. The insights from these models can be succinctly represented on a van Krevelen diagram, as depicted in Fig. 6.

- When aiming for a high solid yield, lignocellulosic biomass with an H/C ratio greater than 1.5 and biodegradable waste with an H/C ratio greater than 1.4 are not recommended for conversion via torrefaction and hydrothermal carbonization, respectively. Additionally, biodegradable waste with an O/C ratio lower than 0.4 is ideally treated using HTC to yield hydrochar with a high HHV.
- Biodegradable waste and biomass possessing an O/C ratio above 0.8 could be converted using HTL to achieve a substantial yield of heavy organics.
- HTG processing of coal with an O/C ratio below approximately 0.10 tends to favor the production of CO and $CH_4$. However, when the O/C ratio exceeds 0.1, the production of $H_2$ and $CO_2$ is more prevalent.
- For biomass with an O/C ratio less than approximately 0.6, pyrolysis is the recommended treatment to yield a high quantity of bio-oil. Conversely, biomass with an O/C ratio ranging from about 0.6 to 0.7 has the potential to produce bio-oil with increased viscosity.
- Biomass exhibiting a H/C ratio of less than approximately 1.5 can undergo pyrolysis to produce superior-quality pyro-gas, characterized by a high $CO/CO_2$ ratio.
- The $CH_4$ content in gases stemming from biomass gasification can be augmented when the biomass's O/C is less than 0.4. Conversely, a higher CO content is achieved when the biomass's O/C exceeds 0.5.

## Conclusions

We have constructed a series of van Krevelen diagrams to visually illustrate the relationships between feedstocks and products in thermal conversion techniques including torrefaction, hydrothermal carbonization, hydrothermal liquefaction, hydrothermal gasification, pyrolysis, and gasification. The reliability of these diagrams is evaluated based on the model's accuracy and the significance of the H/C and O/C parameters. Interestingly, the diagrams exhibit a peculiar pattern associated with low-reliability performance, particularly in the case of pyrolysis-char diagrams.

These diagrams contribute to a better understanding of the respective thermal processes and provide valuable insights for decision-making in practical scenarios. Specifically, they assist in the selection of an appropriate thermal treatment method for a specific feedstock, thereby optimizing the overall performance of the thermal process by considering the blending of different feedstocks to achieve optimal H/C and O/C ratios. By utilizing these diagrams, stakeholders can make informed choices and maximize the efficiency of thermal conversion processes.

In addition to the feedstock-product relationship, the feedstock-reaction connection is also of great importance. The same methodology can be applied to create diagrams that illustrate the relationship between the feedstock and pyrolytic activation energy (Supplementary Note 5). Therefore, further investigation to establish a series of diagrams expressing the feedstock-reaction relationship is of significant interest.

## Methods

The present study builds upon the databases, methods, and findings of eight previous works that specifically investigated ML in the context of torrefaction[70], hydrothermal carbonization (HTC)[71], hydrothermal liquefaction (HTL)[55], hydrothermal gasification (HTG)[72], pyrolysis[36,73,74], and gasification[75]. The referred works and their input and output parameters are given in Table 1.

**Table 1 Summary of referred previous works.**

| Process | Feedstock | Product | Input parameter | Output parameter |
|---|---|---|---|---|
| Torrefaction | Lignocellulosic biomass | Torrefied biomass[70] | C, H, N, ash, VM, size, time, T, $CO_2$, $O_2$ | Yield |
| Hydrothermal carbonization | Biodegradable waste | Hydrochar[71] | C, H, N, O, FC, ash, VM, time, T, WC | Yield, C, H/C, O/C, N/C, CR, ER, HHV |
| Hydrothermal liquefaction | Biodegradable waste & biomass | Bio-oil[55] | C, H, N, O, S, ash, protein, lipid, carbohydrate, time, T, extraction order | Yield, N, ER |
| Hydrothermal gasification | Coal | Gas[72] | C, H, N, O, S, FC, ash, VM, moisture, time, T, ER, CON | $H_2$, CO, $CH_4$, $CO_2$ |
| Pyrolysis | Lignocellulosic Biomass | Char[73] | C, H, N, O, FC, ash, VM, time, T, HR | Yield, C, H, N, O, FC, ash, VM |
| Pyrolysis | Biomass | Bio-oil[36] | C, H, N, O, FC, ash, VM, size, T, HR, room temperature | Yield, H/C, O/C, HHV, viscosity |
| Pyrolysis | Biomass | Gas[74] | C, H, N, O, FC, ash, VM, size, T, HR, FR | Yield, CO, $CO_2$, $H_2$, $CH_4$ |
| Gasification | Biomass & plastics | Gas[75] | C, H, O, ash, moisture, ER, T, steam/biomass ratio, bed material | Yield, $H_2$, $CH_4$, CO, $CO_2$ |

*VM volatile matter, T temperature, FC fix carbon, WC water content of HTC process, CR carbon recovery, ER energy recovery, HHV higher heating value, CON concentrator, HR heating rate, ER equivalent rate, CON concentrator, HR heating rate, FR flow rate.*

Mass yield (Yield) is typically determined directly using a well-established equation:

$$\text{Yield} = \frac{m_{\text{dry product}}}{m_{\text{dry feedstock}}} \quad (1)$$

The energy recovery (ER) is defined as follows[44]:

$$\text{ER} = \text{Yield} \cdot \frac{\text{HHV}_{\text{product}}}{\text{HHV}_{\text{feedstock}}} \quad (2)$$

The carbon recovery (CR) has been determined using the following equation[71]:

$$\text{CR} = Y_m \cdot \frac{\text{Canbon content}_{\text{product}}}{\text{Carbon content}_{\text{feedstock}}} \quad (3)$$

These previous works have demonstrated the feasibility and reliability of utilizing the datasets they employed to develop robust ML models for the respective thermal conversion processes. By leveraging the insights and findings from these prior studies, we have been able to enhance our understanding and analysis of thermal conversion processes through the application of ML techniques. Specifically in this study, we have engineered some features and targets for a better conclusion. These changes include:

1. Converting the contents of H and O to the atomic ratios of H/C and O/C, respectively.
2. Some minor revisions of the datasets where the former researchers made some mistakes.
3. For the pyrolysis-gas models, we investigate the outputs of yield and the atomic ratio of $CO/CO_2$.

The original datasets and their preliminary analysis can be obtained from the sources cited in the respective papers or can be requested from the authors directly. These datasets serve as a foundation for our work and have been modified to improve the consistency and accuracy of our analyses.

Among the eight referenced studies, the random forest method has been implemented most frequently, with a testing $R^2$ value greater than 0.75 (as shown in Fig. 1). To establish a set of general diagrams for different thermal conversion processes, we consistently employ the random forest regressor for generation of the 3D van Krevelen diagrams.

To generate the van Krevelen diagram, a two-way partial dependence analysis is performed based on the constructed model, focusing on the H/C and O/C ratios. The reliability of the produced diagram is highest within the ranges where the training data is most abundant. Therefore, appropriate ranges for H/C and O/C are determined by evaluating the kernel density of the training data distribution (Supplementary Note 7).

During the analysis, all other input parameters are set to their mean values from the training dataset. The resulting two-way partial dependence plots represent the predicted outcomes under specific input conditions. To obtain a van Krevelen diagram that can represent the relationship between feedstock and product more generally, smoothing is applied to the original two-way partial dependence plots (Supplementary Fig. 5).

## Data availability
The data that support the findings of this study are available from the corresponding author upon reasonable request. The information on the dataset used for training biomass torrefaction machine learning model is given in Supplementary Data 1.

## Code availability
In the conduct of this study, we utilized the Random Forest algorithm as implemented in the Scikit-Learn Python library, version 0.23.1. We would like to acknowledge the developers of Scikit-Learn for providing this valuable resource openly. The algorithm was used in its original form without any modifications, ensuring the reproducibility and integrity of our research. For further reference, the library can be accessed at https://scikit-learn.org/0.23/.

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

## Acknowledgements

The financial support of the foundation of Jiangsu Key Lab of Biomass Energy and Material (JSBEM-S-202101) is appreciated. S.W. would like to thank the support of the Research Fund for High-level Talents Introduction of Nanjing Forestry University. Y.W. would like to thank the financial support from (1) the European Commission and the Swedish Research Council Formas for funding in the frame of the collaborative international consortium (RECOWATDIG) financed under the 2018 Joint call of the

WaterWorks2017 ERA-NET Cofund and (2) the National Research Foundation, Singapore, and A*STAR under its Low-Carbon Energy Research (LCER) Funding Initiative (FI) Project (U2102d2011, WBS: A-8000278-00-00).

## Author contributions

Conceptualization: Y.Wen. Methodology: S.W., Z.S., Y.Wen. Software: S.W., Z.S., Y.Wen. Formal analysis: Y.Wang, Y.Wen. Investigation: S.W., Z.S., Y.Wen. Resources: T.O., N.T. Writing—original draft: S.W. (pyro-gas), Z.S. (pyro-char), Y.Wen (overall), L.N. (HTC and torrefaction), R.P. (pyro-bio-oil), Y.X. (HTL & HTG), I.N.Z. (gasification), K.J. (torrefaction), C.A.-B. (HTC). Writing—review & editing: S.W., Y.Wen, L.N., I.N.Z., K.J., C.A.-B., T.O., P.G.J., W.Y. Visualization: Y.Wang, Y.Wen, C.T. Supervision: H.P.-K., P.G.J., W.Y., K.S., J.J., S.K., C.-H.W. Project administration: Y.Wen. Funding acquisition: H.P.-K., P.G.J., W.Y., K.S., J.J., S.K., C.-H.W.

## Competing interests

The authors declare no competing interests.
