## [Peer Review File · Communications Chemistry]

Reviewers' comments:

Reviewer #1 (Remarks to the Author):

The submitted manuscript describes the relationships between feedstock properties and products from thermal conversion processes. This topic is of broad scientific interest. The study demonstrates the use of machine learning to support the visualization of van Krevelen diagrams. This is an interesting combination. The article's merit is enhanced by the scope of the analysis, which includes several different conversion pathways. The results show some clear trends between biomass properties (H/C and O/C ratios) and the product yields along with other properties. The article may be published. There are a few issues for the authors to consider.

Several studies show that ash content is a key driver of thermochemical conversion processes. Ash content is among the feedstock properties collected. However, the text does not mention ash content or its potential impacts on the conversion processes. This is a major omission. A couple of items that should be considered to address this are:

1. What was the range of ash content in the feedstock samples?
2. Did ash have any general impact on any or all thermochemical processes as suggested in the literature?

Machine learning models yield black-box models that are difficult to understand. The article could have used explainable machine learning to provide more insight into the models' decisions. Alternatively, simplified regressions of the results could help scientists verify known patterns or identify novel relationships.

The article suggests general explanations for the observed phenomena, but these are not supported by references. For example, the statement below would benefit from a reference:

Line 150: ... a higher 150 hemicellulose content results in more intense devolatilization and, ultimately, a lower solid yield from the process for the same severity of the torrefaction process

The data employed by the study is not provided. This limits the reproducibility of the study. The SI indicates that most of the data is publicly available, not including the torrefaction results. It would have helped the scientific community if the remaining data had been made publicly available.

Reviewer #2 (Remarks to the Author):

van Krevelen diagrams were originally used when studying coal, but have become more widely used in later times in association with research into biomass, development of sustainable energy such as the use of biofuels, and environmental studies such as the characterization of natural organic matter and dissolved organic matter. The manuscript is well written and provides a good introduction to usage of

van Krevelen diagrams. The authors seek to build upon this through the use of machine learning to understand different processes, particularly with respect to feedstocks and products. The work will have a reasonably broad interest, but is currently lacking in finer detail in areas relating to the data sets and data processing approaches.

General comments:

- A variety of data processing methods is used and are not explained, but it would help the reader to follow the work if the different terms and techniques were explained. The Pearson correlation, Kernel density diagram, random forest, and more are terms that will be understood by those working in data science, but will be less well understood for a broader readership.
- It would be useful for the authors to differentiate this work from preceding work on interpretation of van Krevelen diagrams, including some of the original work by van Krevelen himself. Following different reaction pathways has been covered previously but is a significant part of the manuscript, and so the focus here should be the machine learning aspects and what new information these can offer.
- There should be much greater detail of the data processing methods used here because this is the most significant part of the work, followed by the resulting conclusions. In order to follow the work, assess the validity of the conclusions, and make recommendations for further developments, this should be provided for the readers.

Figure 1: The graphics are low resolution and it can be difficult to read the text. I would suggest that the authors create a higher resolution version or that they enlarge the labels on the various graphs.

Figure 2: This representation looks unusual because of the continuous nature. There will be a limited number of possible chemical compositions and these can be found at intersections for different H/C and O/C ratios, which give rise to the familiar appearance. It is more usual to see data markers instead of gradients.

Figure 3: Similarly to Figure 2, the appearances are unusual. The plot titles should also be explained. For example, seeing "H/C" or "O/C" above plots of H/C vs. O/C is unclear because those parameters are already found within the plots, but the two graphs are different.

Figure 4: I'd recommend choosing colors other than red and green because these plots may be harder to follow for those with "color blindness."

Data availability: As the entire paper is based upon use of existing data sets and development of data processing approaches, showing none of the raw data, algorithm development, nor actual code is something which cannot be overlooked. Stating that these can be provided upon reasonable request will leave many readers unsatisfied. Even if not all of these are covered, there should be at least some "raw" element included. The work is somewhat opaque if none of this is included and it becomes harder to recommend publication.

Reviewer #3 (Remarks to the Author):

1. Fig. 1: please show figures from left to right, and values shown in figures must be written in larger font size. In all other figures, numbers cannot be read.
2. Introduction can discuss the similar work, and authors need to clearly state the novelty of the present work.
3. Fig. 3- HTC- figure for C: when C is higher (dark blue), why do we see a higher H/C value as well?
4. What's missing in this model, is the effect of reactor type and operating conditions such as temperature and heating rate and residence time, etc. They must be mentioned in the manuscript, and the model must be limited to the conditions considered.
5. How many datapoints were used to develop this model? One study for each process is definitely not enough for a ML model, especially when big claims are made.
6. Generalisation should be avoided only by relying on one study. For example: "an increased O/C ratio in biomass tends to enhance the H/C and O/C ratios in bio-oil." How does a higher oxygen in biomass can result in a higher H in the resulting bio-oil?
7. "It can be concluded that bio-oils derived from biomass with high H/C and O/C ratios have more superior fuel properties." This statement is not accurate, as this may be correct only for viscosity.
8. Fig. 4- HTL bio-oil, figure O/C: feedstock with higher O/C produces an oxygenated bio-oil with a high O/C, hence a lower HHV. But in figure HHV, it's seen that higher O/C in feedstock offers a higher HHV in bio-oil which is unrealistic. Also, in this figure, let's consider an extreme case of a feedstock with no oxygen and O/C of zero. I would personally expect to see a relatively high HHV, but the figure shows the lowest HHV.
9. At the end, I would expect to see a discussion section where authors recommend a process for a particular mixed feedstock. For instance, if a feedstock has O/C and H/C of 0.7 and 1.5, what would be the preferred thermochemical treatment to obtain a product with the lowest O/C, etc.

Reviewers' comments:

Reviewer #1 (Remarks to the Author):

The submitted manuscript describes the relationships between feedstock properties and products from thermal conversion processes. This topic is of broad scientific interest. The study demonstrates the use of machine learning to support the visualization of van Krevelen diagrams. This is an interesting combination. The article's merit is enhanced by the scope of the analysis, which includes several different conversion pathways. The results show some clear trends between biomass properties (H/C and O/C ratios) and the product yields along with other properties. The article may be published. There are a few issues for the authors to consider.

Several studies show that ash content is a key driver of thermochemical conversion processes. Ash content is among the feedstock properties collected. However, the text does not mention ash content or its potential impacts on the conversion processes. This is a major omission.

Re: Thank you for your constructive feedback and positive remarks on our manuscript. We genuinely appreciate your recognition of the novelty and breadth of our study. We concur with your observation regarding the significance of ash content in thermochemical conversion processes. In our current study, the primary emphasis was on harnessing machine learning to enhance the visualization of the van Krevelen diagram, which predominantly focuses on the H/C and O/C ratios of feedstocks. Consequently, the effects of ash content were not explored in depth. To address your valid concerns, we have incorporated a supplementary note, titled "Catalytic Effect of Ash on Thermal Conversion," in the supplementary file. This note aims to elucidate the role of ash in influencing the thermal conversion processes of organic feedstocks.

We are grateful for your suggestion and will certainly consider a more comprehensive exploration of ash content's impact in our subsequent research endeavors.

A couple of items that should be considered to address this are:

Q1. What was the range of ash content in the feedstock samples?

Re: Thank you for your insightful question. To address this, we gathered data from various databases, leading to a range of ash content values for different models. We've prepared a box plot to provide a clear visual representation of this range. You can find this plot in Supplementary Note 2. We believe this addition will offer a clearer understanding of the ash content variability in our study.

Supplementary Figure 2. The ash content in the feedstock in the database for each model.

Q2. Did ash have any general impact on any or all thermochemical processes as suggested in the literature?

Re: Thank you for raising this important point. Indeed, the impact of ash on thermochemical processes is contingent upon its specific composition. Ash can act as a catalyst during biomass thermal conversion, and its effect can vary based on its constituents. For a comprehensive understanding, we have provided a detailed description in Supplementary Note 2.

Q3. Machine learning models yield black-box models that are difficult to understand. The article could have used explainable machine learning to provide more insight into the models' decisions. Alternatively, simplified regressions of the results could help scientists verify known patterns or identify novel relationships.

Reply: Thank you for highlighting the importance of model interpretability in machine learning. We concur that explainable machine learning is gaining significant traction in the research community. In our study, we employed the Random Forest algorithm, which, contrary to some perceptions, offers a good degree of interpretability. Specifically, the decisions made by this algorithm can be elucidated using tree-based feature importance analyses. We have provided a comprehensive description of our machine learning model's interpretation in Supplementary Note 9, titled "Fitting results and the interpretation of models". We believe this section addresses the concerns raised and offers insights into the model's decision-making process. We appreciate the suggestion of simplified regressions and will consider it for future work to further enhance the interpretability and verification of our results.

Q4. The article suggests general explanations for the observed phenomena, but these are not supported by references. For example, the statement below would benefit from a reference:

Line 150: ... a higher 150 hemicellulose content results in more intense devolatilization and, ultimately, a lower solid yield from the process for the same severity of the torrefaction process

Reply: Thank you for pointing out the need for references to support the explanations provided in the manuscript. I appreciate your attention to detail. In response to your feedback, we have revisited the manuscript and added pertinent references to substantiate the statement on line 150 regarding the relationship between hemicellulose content and devolatilization intensity. Furthermore, we have identified and addressed other instances in the manuscript where similar issues arose and have ensured that they are now supported by appropriate citations from the literature. We believe that these additions will enhance the rigor and credibility of our work. Thank you for helping us improve the quality of our manuscript.

Q5. The data employed by the study is not provided. This limits the reproducibility of the study. The SI indicates that most of the data is publicly available, not including the torrefaction results. It would have helped the scientific community if the remaining data had been made publicly available.

Reply: Thank you for the comment. Supplementary data, which contains the torrefaction data, is now attached.

Reviewer #2 (Remarks to the Author):

van Krevelen diagrams were originally used when studying coal, but have become more widely used in later times in association with research into biomass, development of sustainable energy such as the use of biofuels, and environmental studies such as the characterization of natural organic matter and dissolved organic matter. The manuscript is well written and provides a good introduction to usage of van Krevelen diagrams. The authors seek to build upon this through the use of machine learning to understand different processes, particularly with respect to feedstocks and products. The work will have a reasonably broad interest, but is currently lacking in finer detail in areas relating to the data sets and data processing approaches.

Reply: Thank you for your insightful remarks and for recognizing the broader applications of van Krevelen diagrams, as well as the potential of our manuscript. We acknowledge the importance of providing a comprehensive understanding of the data sets and data processing approaches. In response to your feedback, we have expanded upon the details related to the data sets and data processing methods. These additional details can now be found in Supplementary Note 4, titled "Dataset Processing Approaches." We believe that this enhancement will provide readers with a clearer understanding of our methodology and its implications. We appreciate your constructive feedback, which has been instrumental in refining our manuscript. We hope that these revisions address your concerns and make our work more accessible and informative to the broader audience.

General comments:

- A variety of data processing methods is used and are not explained, but it would help the reader to follow the work if the different terms and techniques were explained. The Pearson correlation, Kernel density diagram, random forest, and more are terms that will be understood by those working in data science, but will be less well understood for a broader readership.

Reply: Thank you for your valuable feedback emphasizing the importance of making our work accessible to a broader readership. We recognize that while some terms and techniques are commonplace in data science, they might not be as familiar to readers from other disciplines. To address this, we have added Supplementary Notes to provide a more comprehensive understanding of the data processing methods used. Specifically:

Supplementary Note 4 delves into the Pearson correlation coefficient and the random forest regression algorithm.

Supplementary Note 8 provides insights into kernel density analysis.

Supplementary Note 9 elucidates the feature importance method.

We hope that these additions will make the manuscript more reader-friendly and allow those less familiar with data science terminology to better grasp the methodologies employed in our study.

- It would be useful for the authors to differentiate this work from preceding work on interpretation of van Krevelen diagrams, including some of the original work by van Krevelen himself. Following different reaction pathways has been covered previously but is a significant part of the manuscript, and so the focus here should be the machine learning aspects and what

new information these can offer.

Reply: Thank you for the comments. Indeed, a detailed introduction of prior studies serves not only to establish the foundation of the study but also to underscore the contributions of the present research. It is noteworthy that Van Krevelen did not extensively utilize the Van Krevelen diagram, and this tool has garnered more attention in recent decades. Subsequently, three paragraphs in the introduction have been revised to offer a more comprehensive overview of previous studies that employed the Van Krevelen diagram to interpret reactions. This revision also aims to accentuate the novelty of the present study:

“

The van Krevelen diagram, introduced by Dirk Willem van Krevelen in 19508, displays the atomic ratios of H/C and O/C and was originally used to illustrate humification and coal formation processes visually⁹. Over time, it has been recognized as a useful tool for estimating main compound categories and reflecting their calorific values¹⁰. Consequently, its application has expanded beyond coal, denoting relevant properties of diverse materials, including biomass, biodegradable waste, and various chemicals, both pre- and post-reactions¹¹⁻¹⁹.

In the field of thermal conversion, the van Krevelen diagram has been widely used to intuitively indicate differences in H/C and O/C ratios among feedstocks and products in processes such as torrefaction²⁰, hydrothermal carbonization²¹, pyrolysis²², and gasification²³. This application provides a unique way to visually illustrate the directions of not only thermal conversions but also other chemical reactions²⁴⁻²⁸. **However, real feedstocks, such as biomass and biodegradable waste, are typically mixtures, implying that numerous reactions can occur during the thermal conversion process. Consequently, previous investigations using the van Krevelen diagram to understand the directions of several specific reactions can be challenging to apply to the analysis of mixtures.** On the other hand, although there have been studies using the van Krevelen diagram to illustrate the thermal conversion reactions of real biomass and biodegradable wastes such as algae²⁹, lignocellulosic biomass³⁰, and digestate³¹, typically only single or a few cases are reported in each study. **Therefore, there is interest in addressing these gaps and creating van Krevelen diagrams that better reflect the real-world applications of different thermal conversion techniques.**

Machine learning (ML) has become widely used in various fields³², including constructing models for thermal conversion processes^{33,34}. In most of the reported ML studies of thermal processes, the constructed ML model can predict the output from given input parameters with a coefficient of determination (R²) higher than 0.8³³. One ML interpretation method, the partial dependence plot, can be used to evaluate the marginal effects of selected input variables on the output value³⁵. By using the H/C and O/C ratios of feedstocks as input parameters for an ML model and plotting the two-way partial dependence of these input variables on the output value, a three-dimensional van Krevelen diagram can be created. **It will be promising to use the ML method to construct the van Krevelen diagram: using the database yielded from experiments with mixture feedstock will give insight into the corresponding thermal process to treat real feedstock.**

”

- There should be much greater detail of the data processing methods used here because this is the most significant part of the work, followed by the resulting conclusions. In order to follow the work, assess the validity of the conclusions, and make recommendations for further developments, this should be provided for the readers.

Reply: Thank you for emphasizing the importance of providing a thorough description of our data processing methods. We understand that this is a crucial aspect of our work, and it's essential for readers to fully grasp our methodology to evaluate the validity of our conclusions.

In response to your feedback, we have elaborated on the data processing and analysis methods. These comprehensive details can be found in Supplementary Notes 4, 8, and 9. We believe that these additions will offer readers a clearer understanding of our approach and its implications.

We genuinely appreciate your constructive feedback, which has been invaluable in enhancing the clarity and depth of our manuscript.

Figure 1: The graphics are low resolution and it can be difficult to read the text. I would suggest that the authors create a higher resolution version or that they enlarge the labels on the various graphs.

Reply: Thank you for bringing to our attention the issue with the resolution of Figure 1. We understand the importance of clear visuals for effective communication of our findings.

To clarify, the original figure we prepared for the manuscript is of high resolution. It seems that the resolution might have been compromised during the conversion during submission process, leading to the generation of a lower resolution PDF file.

In response to your feedback, we have restructured Figure 1 to enlarge each of the smaller figures contained within it. Figure 1 serves to introduce the process of this study, with each smaller figure acting as a schematic diagram.

Figure 2: This representation looks unusual because of the continuous nature. There will be a limited number of possible chemical compositions and these can be found at intersections for different H/C and O/C ratios, which give rise to the familiar appearance. It is more usual to see data markers instead of gradients.

Reply: Thank you for your comments. Indeed, numerous previous studies have utilized the Van Krevelen diagram to plot points corresponding to different feedstocks and to indicate potential reaction pathways. However, it is important to note that feedstocks in real-world scenarios often consist of mixed substrates, which can cause their positions to vary over time. Consequently, one of the novel aspects of this study is the application of machine learning methods to construct a continuous Van Krevelen diagram. This approach aims to reflect the feedstock-product relationship in a more comprehensive manner. To better illustrate this, the following section has been revised in the Introduction:

“

In the field of thermal conversion, the van Krevelen diagram has been widely used to intuitively indicate differences in H/C and O/C ratios among feedstocks and products in processes such as torrefaction²⁰, hydrothermal carbonization²¹, pyrolysis²², and gasification²³. This application provides a unique way to visually illustrate the directions of not only thermal conversions but also other chemical reactions²⁴⁻²⁸. **However, real feedstocks, such as biomass and biodegradable waste, are typically mixtures, implying that numerous reactions can occur during the thermal conversion process. Consequently, previous investigations using the van Krevelen diagram to understand the directions of several specific reactions can be challenging to apply to the analysis of mixtures.** On the other hand, although there have been studies using the van Krevelen diagram to illustrate the thermal conversion reactions of real

biomass and biodegradable wastes such as algae²⁹, lignocellulosic biomass³⁰, and digestate³¹, typically only single or a few cases are reported in each study. **Therefore, there is interest in addressing these gaps and creating van Krevelen diagrams that better reflect the real-world applications of different thermal conversion techniques.**

Machine learning (ML) has become widely used in various fields³², including constructing models for thermal conversion processes^{33,34}. In most of the reported ML studies of thermal processes, the constructed ML model can predict the output from given input parameters with a coefficient of determination (R2) higher than 0.8³³. One ML interpretation method, the partial dependence plot, can be used to evaluate the marginal effects of selected input variables on the output value³⁵. By using the H/C and O/C ratios of feedstocks as input parameters for an ML model and plotting the two-way partial dependence of these input variables on the output value, a three-dimensional van Krevelen diagram can be created. **It will be promising to use the ML method to construct the van Krevelen diagram: using the database yielded from experiments with mixture feedstock will give insight into the corresponding thermal process to treat real feedstock.**

”

Figure 3: Similarly to Figure 2, the appearances are unusual. The plot titles should also be explained. For example, seeing “H/C” or “O/C” above plots of H/C vs. O/C is unclear because those parameters are already found within the plots, but the two graphs are different.

Reply: Thank you for the valuable suggestion. Indeed, the previous setup could be confusing. To enhance clarity, the plot titles have been revised to be more accurate; for example, “H/C” has been changed to “Hydrochar H/C”. Similar modifications have been implemented in other figures.

Figure 4: I’d recommend choosing colors other than red and green because these plots may be harder to follow for those with “color blindness.”

Reply: Thank you for the kindly comment. The pyrolysis-gas figures in Figure 4 have been changed as follows:

Moreover, the Figure 5 of gasification diagrams are also modified as follows:

Data availability: As the entire paper is based upon use of existing data sets and development of data processing approaches, showing none of the raw data, algorithm development, nor actual code is something which cannot be overlooked. Stating that these can be provided upon reasonable request will leave many readers unsatisfied. Even if not all of these are covered, there should be at least some “raw” element included. The work is somewhat opaque if none of this is included and it becomes harder to recommend publication.

Reply: Thank you for your comment. All the data are available online, except for the torrefaction data. The Supplementary Data, which contains the torrefaction data, is now attached. More details of the machine learning study process are also included, as presented in the newly added Supplementary Note 9.

Reviewer #3 (Remarks to the Author):

1. Fig. 1: please show figures from left to right, and values shown in figures must be written in larger font size. In all other figures, numbers cannot be read.

Reply: Thank you for the comment. In response to your feedback, we have restructured Figure 1 to enlarge each of the smaller figures contained within it. Figure 1 serves to introduce the process of this study, with each smaller figure acting as a schematic diagram.

Additionally, we have modified all other figures to be more compact, enabling each smaller diagram to be enlarged. Moreover, we have also increased the size of all labels and numbers by two increments to enhance readability. We appreciate your suggestion and will ensure this is rectified in the revised submission.

2. Introduction can discuss the similar work, and authors need to clearly state the novelty of the present work.

Reply: Thank you for your comments. Additional works that have applied the van Krevelen diagram to illustrate the properties of chemicals, as well as to indicate reaction pathways, are now mentioned and discussed. The novelty of this study is subsequently emphasized in the following modified paragraphs in the introduction.

“

The van Krevelen diagram, introduced by Dirk Willem van Krevelen in 19508, displays the atomic ratios of H/C and O/C and was originally used to illustrate humification and coal formation processes visually⁹. Over time, it has been recognized as a useful tool for estimating main compound categories and reflecting their calorific values¹⁰. Consequently, its application has expanded beyond coal, denoting relevant properties of diverse materials, including biomass, biodegradable waste, and various chemicals, both pre- and post-reactions¹¹⁻¹⁹.

In the field of thermal conversion, the van Krevelen diagram has been widely used to intuitively indicate differences in H/C and O/C ratios among feedstocks and products in processes such as torrefaction²⁰, hydrothermal carbonization²¹, pyrolysis²², and gasification²³. This application provides a unique way to visually illustrate the directions of not only thermal conversions but also other chemical reactions²⁴⁻²⁸. **However, real feedstocks, such as biomass and biodegradable waste, are typically mixtures, implying that numerous reactions can occur during the thermal conversion process. Consequently, previous investigations using the van Krevelen diagram to understand the directions of several specific reactions can be challenging to apply to the analysis of mixtures.** On the other hand, although there have been studies using the van Krevelen diagram to illustrate the thermal conversion reactions of real biomass and biodegradable wastes such as algae²⁹, lignocellulosic biomass³⁰, and digestate³¹, typically only single or a few cases are reported in each study. **Therefore, there is interest in addressing these gaps and creating van Krevelen diagrams that better reflect the real-world applications of different thermal conversion techniques.**

Machine learning (ML) has become widely used in various fields³², including constructing models for thermal conversion processes^{33,34}. In most of the reported ML studies of thermal processes, the constructed ML model can predict the output from given input parameters with a coefficient of determination (R²) higher than 0.8³³. One ML interpretation method, the partial dependence plot, can be used to evaluate the marginal effects of selected input variables on the output value³⁵. By using the H/C and O/C ratios of feedstocks as input parameters for an ML model and plotting the two-way partial dependence of these input variables on the output value,

a three-dimensional van Krevelen diagram can be created. **It will be promising to use the ML method to construct the van Krevelen diagram: using the database yielded from experiments with mixture feedstock will give insight into the corresponding thermal process to treat real feedstock.**

”

3. Fig. 3- HTC- figure for C: when C is higher (dark blue), why do we see a higher H/C value as well?

Reply: Thank you for the comment. We have added several sentences to discuss this trend as follows:

“

Lignin has a lower H/C ratio (1.14) compared to cellulose (1.67) and hemicellulose (1.60)⁵⁴. Therefore, a lower H/C ratio in lignocellulosic biodegradable waste indicates a relatively higher lignin content. Lignin exhibits better thermal stability than cellulose and hemicellulose, which is reflected in the HTC Yield diagram: the lower the H/C ratio, the lower the yield. **Similarly, the HTC C diagram illustrates that a lower H/C ratio results in lower C content in the hydrochar. The hydrochar produced from lignin has a relatively lower C content than that produced from cellulose and hemicellulose⁵⁵. Hence, feedstock with a higher lignin content will have a lower H/C ratio and produce hydrochar with lower C content.**

”

4. What’s missing in this model, is the effect of reactor type and operating conditions such as temperature and heating rate and residence time, etc. They must be mentioned in the manuscript, and the model must be limited to the conditions considered.

Reply: Thank you for your insightful comments. We concur that process parameters such as reactor type, reaction temperature, heating rate, and residence time are crucial in thermal conversion processes. It's worth noting that our machine learning models assimilated results from various studies, encompassing a range of feedstocks and diverse process parameters. Our core focus in this study was on elucidating the relationship between the feedstock’s H/C and O/C ratios and the properties of the products using partial dependence analysis. During this analysis, other parameters were set to their mean values from the dataset. This means that while our primary emphasis was on the H/C and O/C ratios, other process parameters were indeed factored into our machine learning models.

5. How many datapoints were used to develop this model? One study for each process is definitely not enough for a ML model, especially when big claims are made.

Reply: For each specific process, we sourced our data from corresponding prior machine learning research. It's imperative to note that these precursor studies employed datasets collated from established experimental articles. Furthermore, these machine learning studies have rigorously validated the feasibility of using their datasets to build accurate machine learning models for their respective processes. The dataset can be obtained from these previous works listed in Table 1.

6. Generalisation should be avoided only by relying on one study. For example: “an increased O/C ratio in biomass tends to enhance the H/C and O/C ratios in bio-oil.” How does a higher oxygen in biomass can result in a higher H in the resulting bio-oil?

Reply: Thank you for the valuable comment. The properties of products yielded from thermal conversion processes are controlled by different feedstocks’ properties and process parameters at the same time. Hence, this work intends to investigate the general relationship between feedstock and thermal conversion products.

A new discussion is added in the manuscript to explain how the higher O/C in biomass can result bio-oil with a higher H/C:

“The H/C and O/C diagrams for Pyrolysis-bio-oil highlight that the O/C ratio of the biomass predominantly influences the H/C and O/C ratios in the bio-oil. Generally, an elevation in the biomass's O/C ratio correspondingly increases the H/C and O/C ratios in the bio-oil. Cellulose and hemicellulose exhibit O/C ratios of 0.83 and 0.80, respectively. These values are noticeably higher than lignin's O/C ratio, which stands at 0.35⁵⁴. Notably, bio-oil derived from the pyrolysis of cellulose and hemicellulose (xylan) demonstrated a superior H/C ratio compared to that from lignin pyrolysis⁶⁸. As a consequence, a higher O/C ratio in biomass, indicating a reduced lignin content, translates to a heightened H/C ratio in the resultant bio-oil.”

7. “It can be concluded that bio-oils derived from biomass with high H/C and O/C ratios have more superior fuel properties.”. This statement is not accurate, as this may be correct only for viscosity.

Reply: Thank you for your comment. This sentence is deleted now.

8. Fig. 4- HTL bio-oil, figure O/C: feedstock with higher O/C produces an oxygenated bio-oil with a high O/C, hence a lower HHV. But in figure HHV, it's seen that higher O/C in feedstock offers a higher HHV in bio-oil which is unrealistic. Also, in this figure, let's consider an extreme case of a feedstock with no oxygen and O/C of zero. I would personally expect to see a relatively high HHV, but the figure shows the lowest HHV.

Reply: Thank you for the comment. The dataset we used was derived from the results of biomass pyrolysis; hence, our discussion is necessarily limited to the realm of biomass pyrolysis. Zero or extremely low O/C ratios are typically observed in plastic feedstocks, and the relationship between plastics and their pyrolysis oils might differ.

An additional discussion we've incorporated for the comment 3 elucidates why a higher O/C ratio in feedstock results in an increased HHV of the bio-oil: a higher O/C ratio in the feedstock indicates a relative abundance of cellulose and hemicellulose. This, in turn, leads to a higher H/C content in the bio-oil. Consequently, the HHV of the produced bio-oil increases, which can be attributed to the elevated H/C content of the bio-oil stemming from the heightened O/C content in the feedstock.

9. At the end, I would expect to see a discussion section where authors recommend a process for a particular mixed feedstock. For instance, if a feedstock has O/C and H/C of 0.7 and 1.5, what would be the preferred thermochemical treatment to obtain a product with the lowest O/C, etc.

Reply: Thank you very much on the comment. A new section is added as shown below:

“

Summary of biomass thermal conversions

Figure 6. Summary of recommendations for the thermal conversion of biomass to achieve different targets, illustrated using a Van Krevelen diagram. Only models created from datasets containing biomass feedstock are included.

A brief summary is provided in this section, outlining recommendations for the thermal conversion of biomass with varying H/C and O/C ratios to achieve different target products. Notably, ML models for torrefaction, HTL, pyrolysis-oil, pyrolysis-gas, and gasification were built using datasets from biomass feedstock. The insights from these models can be succinctly represented on a van Krevelen diagram, as depicted in Fig. 6.

- When aiming for a high solid yield, lignocellulosic biomass with an H/C ratio greater than 1.5 and biodegradable waste with an H/C ratio greater than 1.4 are not recommended for conversion via torrefaction and hydrothermal carbonization,

respectively. Additionally, biodegradable waste with an O/C ratio lower than 0.4 is ideally treated using HTC to yield hydrochar with a high HHV.

- Biodegradable waste and biomass possessing an O/C ratio above 0.8 could be converted using HTL to achieve a substantial yield of heavy organics.
- HTG processing of coal with an O/C ratio below approximately 0.10 tends to favor the production of CO and CH₄. However, when the O/C ratio exceeds 0.1, the production of H₂ and CO₂ is more prevalent.
- For biomass with an O/C ratio less than approximately 0.6, pyrolysis is the recommended treatment to yield a high quantity of bio-oil. Conversely, biomass with an O/C ratio ranging from about 0.6 to 0.7 has the potential to produce bio-oil with increased viscosity.
- Biomass exhibiting an H/C ratio less than approximately 1.5 can undergo pyrolysis to produce superior-quality pyro-gas, characterized by a high CO/CO₂ ratio.
- The CH₄ content in gases stemming from biomass gasification can be augmented when the biomass's O/C is less than 0.4. Conversely, a higher CO content is achieved when the biomass's O/C exceeds 0.5.

REVIEWERS' COMMENTS:

Reviewer #1 (Remarks to the Author):

The authors have addressed the reviewers' comments.

Reviewer #3 (Remarks to the Author):

Authors carefully addressed all comments with additional information to support modeling data. However, I still believe this model must be limited to the range of conditions studied.

Thank you.

Reviewer #1 (Remarks to the Author):

The authors have addressed the reviewers' comments.

Response: Thank you for your valuable comments, which helped to improve the manuscript a lot.

Reviewer #3 (Remarks to the Author):

Authors carefully addressed all comments with additional information to support modeling data.

However, I still believe this model must be limited to the range of conditions studied.

Response: Thank you for your valuable comments and suggestions. We agree that the yielded diagrams in this study are only reliable when they are applied to certain conditions. Hence, we have demonstrated their corresponding limited conditions in the manuscript including feedstock, ranges of feedstocks' H/C and O/C values, accuracy, and feature reliability.